# Effect of Apple Polyphenols on the Antioxidant Activity and Structure of Three-Dimensional Printed Processed Cheese

**DOI:** 10.3390/foods12081731

**Published:** 2023-04-21

**Authors:** Yiqiu Deng, Guangsheng Zhao, Kewei Cheng, Chuanchuan Shi, Gongnian Xiao

**Affiliations:** 1Key Laboratory of Agricultural Products Chemical and Biological Processing Technology, Zhejiang University of Science and Technology, Hangzhou 310023, China; 212003817007@zust.edu.cn; 2Hangzhou New Hope Shuangfeng Dairy Products Co., Ltd., Hangzhou 311100, China; 3Hangzhou Institute for Food and Drug Control, Hangzhou 310017, China; 4Panda Dairy Group Co., Ltd., Wenzhou 325400, China

**Keywords:** apple polyphenols, three-dimensional printing, processed cheese, antioxidant activity, structural stability

## Abstract

Additives can influence the processability and quality of three-dimensional (3D)-printed foods. Herein, the effects of apple polyphenols on the antioxidant activity and structure of 3D-printed processed cheese were investigated. The antioxidant activities of processed cheese samples with different contents of apple polyphenols (0%, 0.4%, 0.8%, 1.2%, or 1.6%) were evaluated using 2,2′-azinobis-(3-ethylbenzothiazoline-6-sulfonic acid) (ABTS) and 2,2-di(4-*tert*-octylphenyl)-1-picrylhydrazyl (DPPH) assays. In addition, the rheological properties and structural characteristics of the processed cheeses were investigated using rheometry, Fourier transform infrared spectroscopy, and fluorescence spectroscopy. Then, the final printed products were analyzed for comparative molding effects and dimensional characteristics. it was found that apple polyphenols can significantly improve the antioxidant activity of processed cheese. When the amount of apple polyphenols added was 0.8%, the 3D shaping effect was optimal with a porosity rate of 4.1%. Apple polyphenols can be used as a good antioxidant additive, and the moderate addition of apple polyphenols can effectively improve the antioxidant and structural stability of 3D-printed processed cheese.

## 1. Introduction

Three-dimensional (3D)-printed food, which can be digitally designed, has an important role in the field of food processing and production [1]. The increasing demand for functional foods has greatly contributed to new 3D printing applications in the food industry [2]. Hot-extrusion printing, which is currently the most widely used 3D printing technology in the food sector, has good compatibility with traditional food materials [3]. Compared with traditional food manufacturing processes, 3D printing technology can meet the demand for personalized customized functional food by preparing food ink with functional active ingredients or additives added to the raw materials [4].

Cheese, as a nutritious dairy product, can be roughly divided into two categories: natural cheese and processed cheese. Processed cheese is a homogeneous and easy-to-store dairy product made by heating and stirring different natural cheeses with added emulsifying salts and other dairy or non-dairy ingredients [5]. Compared with natural cheese, processed cheese is easier to store, more delicious, and can effectively improve the functional and structural characteristics of raw milk cheese by adding food additives [6]. With the continuous growth of cheese consumption in China and in response to the demand for healthy functional food from consumers, the combination of processed cheese and 3D printing technology can provide new ideas for expanding new and high-value functional processed cheese products.

Polyphenols are secondary metabolites of plants and have advantages such as antioxidant and antibacterial properties, which can endow food with good functional characteristics [7]. Among them, apple polyphenols are a common polyphenolic compound with strong antioxidant properties. Studies have shown that the antioxidant properties of apple polyphenols can effectively improve the function of the liver and intestines and prevent liver and digestive system diseases. For example, Huang et al. [8] added apple polyphenols to pig feed, which activated the Nrf2/Keap1 signaling pathway, improved the antioxidant capacity of the pig intestine, and effectively improved the mechanical and immune barriers of the intestine. This can provide effective references for the development of high-value functional 3D-printed processed cheese.

Related research reports that apple polyphenols can also form complex gels with protein macromolecules, but their structure may be affected by apple polyphenols, thereby affecting the properties and functions of the gel [9]. For example, Zhou et al. [10] found that apple polyphenols have a significant impact on the gel properties of whey protein isolate, and moderate addition of apple polyphenols can effectively improve its structural characteristics. It is worth noting that the gel properties of food are also an important factor affecting the 3D printing process. Therefore, while fully utilizing the antioxidant advantages of apple polyphenols, it is also necessary to further understand the influence of apple polyphenols on the gel structure of food and its application value.

Based on 3D printing technology and using Anjia cream cheese as the main raw material, this experiment intends to use apple polyphenols to improve processed cheese and develop high-antioxidant 3D-printed processed cheese, exploring the effects of apple polyphenols on the antioxidant and structural stability of 3D-printed processed cheese, and providing effective references for the production and application of functional 3D-printed processed cheese.

## 2. Materials and Methods

### 2.1. Reagents

Apple polyphenols (75% purity; food grade) were obtained from Shanghai Yuanye Biological Co., Ltd. Anjia cream cheese (15% protein, 57% fat, 1% carbohydrate, and 14% sodium; food grade) was obtained from Shanghai Ever Natural Trading Co., Ltd. (Shanghai, China), TG (food grade) was obtained from Hefei Bomei Biotechnology Co., Ltd. (Hefei, China), Emulsifying salts (sodium polyphosphate and sodium pyrophosphate; food grade) were obtained from Zhejiang Noyi Biotechnology Co., Ltd. (Haining, China), Phosphate buffer solution (PBS) was obtained from Fly Clean Biotechnology Co., Ltd. (Shanghai, China), Ethanol and KBr were obtained from Shanghai Lingfeng Chemical Reagent Co., Ltd. (Shanghai, China), 2,2′-azinobis-(3-ethylbenzothiazoline-6-sulfonic acid) (ABTS) and 2,2-di(4-tert-octylphenyl)-1-picrylhydrazyl (DPPH) were obtained from Sangon Biotech Co., Ltd. (Shanghai, China).

### 2.2. Sample Preparation

Anjia cream cheese (50 g) was mixed with 1.5% emulsifying salts (sodium pyrophosphate and sodium polyphosphate, 1:1, *v*/*v*) and 25% distilled water in a constant temperature water bath at 80 °C for 20 min. The mixture was placed under a high-speed homogenizer and homogenized intermittently at 6500 rpm for 2 min until uniform and non-grainy in appearance. After cooling to room temperature, 0.4% TG was added and then different amounts of apple polyphenols were added (0%, 0.4%, 0.8%, 1.2%, or 1.6%). The mixture was placed in a constant temperature water bath at 55 °C for 30 min, sterilized at 85 °C for 30 min, cooled, refrigerated at 4 °C for 24 h, and then used for 3D printing and sample analysis.

### 2.3. Determination of Antioxidant Activity

#### 2.3.1. Preparation of Sample Solution

pH 7.4, 10 mmol/L phosphate buffer solution was prepared and used as a solvent to prepare 1 mg/mL of processed cheese stock solution. This was refrigerated at 4 °C and used as the sample solution for subsequent determination of antioxidant capacity.

#### 2.3.2. 2,2-Di(4-tert-octylphenyl)-1-picrylhydrazyl (DPPH) Assay

First, a 0.06 mmol/L DPPH solution was prepared. A blank was prepared by mixing 4 mL of the DPPH solution with 0.2 mL of ethanol. The initial absorbance of the blank (A_0_) was measured at 515 nm. The test solution was prepared by mixing 4 mL of the DPPH solution with 0.2 mL of sample. After incubating at room temperature and protecting from light for 1 h, the absorbance of the test solution (A_p_) was measured at 515 nm. A control solution was prepared by mixing 4 mL of ethanol with 0.2 mL of sample, and the absorbance of this solution (A_c_) was measured at 515 nm [11]. Parallel experiments were performed three times. The DPPH free radical scavenging rate was calculated as follows:DPPH free radical scavenging rate (%) = (A_0_ − A_p_ − A_c_)/A_0_ × 100%(1)

#### 2.3.3. 2,2′-Azinobis-(3-ethylbenzothiazoline-6-sulfonic Acid) (ABTS) Assay

A 7 mmol/L ABTS free radical solution and a 2.45 mmol/L potassium persulfate solution were prepared and mixed in equal volumes. The mixture was placed in the dark for 12 h and then diluted with anhydrous ethanol to obtain an ABTS working solution with an absorbance of 0.70 ± 0.02 at 732 nm. A 1.0 mL sample solution of processed cheese with different apple polyphenol contents of 1.0 mg/mL was mixed with 3.9 mL of the ABTS working solution and reacted for 6 min at room temperature. Anhydrous ethanol was used as a blank instead of the sample solution, and the absorbance of the reaction sample was measured at 734 nm [11]. Parallel experiments were performed three times. The ABTS free radical scavenging rate was calculated as follows:ABTS free radical scavenging rate (%) = (1 − A_1_/A_0_) × 100(2)
where A_1_ is the absorbance value of the processed cheese sample after reacting with ABTS, and A_0_ is the absorbance value of anhydrous ethanol instead of the sample solution after reacting with ABTS.

### 2.4. Determination of Rheological Characteristics

Rheological characterization was performed according to the method of Huang et al. [12] with slight modifications. The dynamic rheological properties were evaluated using a rheometer (DHR-2, TA Instruments, New Castle, DE, USA) to determine the effect of different apple polyphenol contents on the rheological characteristics of the cheese samples. A 40 mm diameter plate was used with a gap of 1 mm. The samples were placed on the platform for 1 min to warm up to 25 °C and then subjected to dynamic frequency scanning at angular frequencies of 0.1–100 rad/s. In addition, to evaluate the printability of the processed cheese samples, apparent viscosity profiles were collected. The apparent viscosity was recorded as the shear rate was scanned in the range of 0.1–100 s^−1^ at 25 °C.

### 2.5. Determination of Structural Properties

#### 2.5.1. Fourier Transform Infrared (FT-IR) Spectroscopy

Processed cheese samples with different amounts of apple polyphenols were placed in a freezer at −80 °C for 24 h. After freeze-drying, the processed cheese samples were ground into powder and mixed with KBr (1:100, *v*/*v*). FT-IR spectra were recorded at room temperature using an FT-IR spectrometer (V70 IR Spectrometer, Bruker, Mannheim, Germany) [13].

#### 2.5.2. Endogenous Tryptophan Fluorescence

The fluorescence spectra of the processed cheese samples with different apple polyphenol contents were measured using a fluorescence spectrometer (F4500, Hitachi, Tokyo, Japan) according to the method of Geng et al. [14]. The samples were diluted to 0.1 mg/mL using 0.5 mol/L phosphate buffer solution (PBS) and then placed in a cuvette. Fluorescence spectra were recorded in the range of 280–400 nm using an excitation wavelength of 295 nm, an excitation spacing of 5 nm, an emission spacing of 5 nm, and a data acquisition rate of 500 nm/min.

#### 2.5.3. Scanning Electron Microscopy

A scanning electron microscope (SU1510, Hitachi, Japan) was used to observe the microstructure of the processed cheese samples with different apple polyphenol contents. The samples were freeze-dried at −80 °C for 24 h, fixed on a copper plate, and then sprayed with gold. Scanning electron micrographs were obtained at a magnification of 1000× and an accelerating voltage of 15 kV [15].

### 2.6. Measurement of 3D Printing Molding Effects

To evaluate the 3D printing and molding performance of the processed cheese samples, a food 3D printer (FOODBOT-S2, Hangzhou Shiyin Technology Co., Ltd., Hangzhou, China) was used and a geometric model of a horse was selected from the food 3D printer gallery. The model volume was (48 mm × 28 mm × 3 mm). The 3D printer parameters were as follows: nozzle diameter, 0.8 mm; nozzle temperature, 25 °C; fill rate, 10%; and print rate, standard. The size features of the printed product were measured using Vernier calipers, and the porosity of samples was determined using Image J software. Parallel experiments were performed three times [16].

### 2.7. Color Analysis

The L*, a*, and b* values of the finished product were measured using a spectrophotometer to analyze its color. The L* value represents brightness, with higher values indicating a whiter sample; the a* value represents the red-green degree, with higher values indicating a more red sample; the b* value represents the yellow-blue degree, with higher values indicating a deeper yellow color in the sample [16].

### 2.8. Sensory Evaluation

According to the method of Lao et al. [17], sensory evaluation was conducted on cheese samples with apple polyphenols added. Sensory evaluation indicators were divided into: color (0–3 points), appearance (0–3 points), flavor (0–4 points), and a total of 10 points. If the cumulative score exceeds 7 points, the overall acceptability of the product is considered high, and the average score of all people is used as the final score of the product.

### 2.9. Statistical Analysis

The statistical analysis was performed using SPSS software (version 26.0, IBMCorp., Armonk, NY, USA). The data were collated and plotted using Origin 9.1. software.

## 3. Results and Discussion

### 3.1. DPPH and ABTS Free Radical Scavenging Rates

The antioxidant activities of the processed cheese samples with added apple polyphenols were evaluated by both DPPH and ABTS methods. As shown in Figure 1, the DPPH free radical scavenging rate increased gradually with the addition of apple polyphenols. The DPPH free radical scavenging rate was positively correlated to the polyphenol content, and the highest scavenging rate of 71.9% was achieved at an apple polyphenol content of 1.6%. When the content of apple polyphenols is greater than 0.4%, the DHHP free radical scavenging rate is above 70%. Thus, apple polyphenols can effectively improve the antioxidant activity of processed cheese. The free radical scavenging activity of polyphenols is attributable to their free phenolic hydroxyl groups, which have a strong ability to provide protons that can reduce highly oxidative free radicals and terminate free radical chain reactions [18].

The ABTS free radical scavenging rates of the processed cheese samples were also positively correlated to the amount of added apple polyphenols (Figure 1). The differences in the radical scavenging rates of ABTS and DPPH are due to differences in solubility. As ABTS is water-soluble, it is more suitable for the determination of water-soluble substances, whereas DPPH is soluble in alcohols and is thus more suitable for the determination of alcohol-soluble substances [19]. The highest ABTS free radical scavenging rate of 76.8% was achieved at an apple polyphenol content of 1.6%. When the content of apple polyphenols is greater than 0.4%, the ABTS free radical scavenging rate is above 60%. These results confirm that the addition of apple polyphenols can effectively improve the antioxidant activity of processed cheese.

### 3.2. Rheological Analysis

The rheological characteristics of the processed cheese samples with added apple polyphenols were investigated. As shown in Figure 2A, the apparent viscosity of all the processed cheese samples gradually decreased with increasing shear rate. Thus, irrespective of the apple polyphenol content, the processed cheese samples exhibited shear thinning, which indicates that the gelation properties are suitable for food 3D printing [20]. This behavior is due to the main phenolic substances in apple polyphenols, including chlorogenic acid and proanthocyanidins, which bind to proteins through hydrogen bonding and hydrophobic interactions to form aggregates [21]. As the content of apple polyphenols increased, the apparent viscosity gradually decreased, which may be due to the interactions between polyphenols and proteins reducing the viscosity of the processed cheese gel system. Notably, the viscosity of a material has an important influence on its 3D printing properties.

Figure 2B shows the energy storage modulus (G′), which is indicative of the elasticity of the protein gel network structure, whereas Figure 2C shows the loss modulus (G″, also known as the viscosity modulus), which provides a measure of gel viscosity [22]. As shown in Figure 2B, when the apple polyphenol content increases from 0% to 1.6%, the G′ values of the processed cheese samples first increased and then decreased as the apple polyphenol content increased, with the maximum value observed at an apple polyphenol content of 0.8%. Thus, within a certain content range, the addition of apple polyphenols promoted the formation of a protein gel network structure and improved the elasticity of processed cheese, but excess polyphenols caused the elasticity of the protein gel system to decrease. As shown in Figure 2C, the G″ values of the processed cheese samples tended to decrease as the apple polyphenol content increased. However, when the apple polyphenol content was 0.8%, 1.2%, and 1.6%, the G′ values were always greater than the G″ values, indicating that apple polyphenols promote the formation of a cross-linked protein structure that provides enhanced elasticity.

### 3.3. Structural Analysis

#### 3.3.1. Scanning Electron Microscopy

The addition of apple polyphenols significantly affected the microstructure of processed cheese (Figure 3). The gel network structure of processed cheese without added apple polyphenols was loose with obvious surface folds and cavities (Figure 3A). With the addition of apple polyphenols, the protein network structure gradually became tighter. Compared with the sample without apple polyphenols (Figure 3A), the samples with added apple polyphenols had flatter and denser structures with some protein cross-linking (Figure 3B–E). Previous studies on protein–polyphenol interactions found that the presence of apple polyphenols enhances the tight binding of protein gels [23], suggesting that the added polyphenols act as a protein cross-linker and thus promote the formation of a cheese gel structure with enhanced mechanical strength.

#### 3.3.2. FT-IR Spectroscopy

The effect of apple polyphenol addition on the structural characteristics of processed cheese was further investigated using FT-IR spectroscopy. As shown in Figure 4A, the FT-IR spectra did not change significantly after the addition of apple polyphenols, indicating that no new functional groups were generated and that the polyphenols were bound to the proteins through non-covalent bonds. The intensities of various characteristic peaks varied depending on the apple polyphenol content, possibly due to changes in the secondary structure of the protein [24]. One of the most useful spectral bands for the analysis of protein secondary structure and conformation is the amide I band (1600–1700 cm^−1^) due to its sensitivity to hydrogen bonding patterns and dipole–dipole interactions [25,26]. Peaks corresponding to various secondary structures, including β-folds (1611–1640 cm^−1^), random curls (1642–1650 cm^−1^), α-helices (1654–1662 cm^−1^), and β-turns (1665–1693 cm^−1^), are located within the spectral range of the amide I band. Therefore, to clarify the effect of apple polyphenols on the secondary structure of processed cheese proteins, PeakFit software was used to calculate the relative percentages of various secondary structures based on the integrated areas of their peaks.

The changes in the gelation properties and strength of the processed cheese samples were related to the changes in the relative content of each protein secondary structure. A higher α-helix content results in a more rigid and tight protein structure, whereas a higher β-fold content results in a more ordered and stable protein structure [27]. As shown in Figure 4B, at apple polyphenol contents of 0.4% and 0.8%, the random curl content decreased, but the α-helix and β-fold contents increased to some extent, indicating a protective effect on the protein gel structure [28], which is consistent with the scanning electron microscopy results (Figure 3A–E). Upon further increasing the apple polyphenol content (>0.8%), the α-helix content gradually decreased, which may be due to excess polyphenols hindering the formation of hydrogen bonds in the protein gel network while promoting protein unfolding and the conversion of α-helices to β-turns [29]. Therefore, the addition of an appropriate amount of apple polyphenols to processed cheese could lead to improved structural stability.

#### 3.3.3. Fluorescence Spectroscopy

Endogenous tryptophan fluorescence is often used to reveal protein conformation. When a protein is in a folded state, tryptophan residues are present in hydrophobic structures within the protein, resulting in relatively high fluorescence intensity. In contrast, when a protein is partially or completely unfolded, the tryptophan residues are more exposed to the protein surface, resulting in decreased fluorescence intensity [30]. As shown in Figure 5, the fluorescence of the proteins in the processed cheese samples group was gradually quenched as the amount of added apple polyphenols increased. This behavior indicates that apple polyphenols induce protein unfolding in processed cheese, thus exposing more tryptophan residues and reducing the fluorescence intensity, which is consistent with FT-IR spectroscopy results.

### 3.4. D Printing Performance and Colour Analysis

To assess the 3D printing performance of the processed cheese samples with added apple polyphenols, 3D printing was carried out using a horse model. As shown in Table 1, after the addition of apple polyphenols, the best 3D printing effect was achieved at an apple polyphenol content of 0.8%, in which the porosity was 4.1%. In this case, the sample was well extruded. This sample exhibited closely stacked layers, no obvious collapse, a smooth surface, and good deformation resistance with no obvious deviations from the blank group sample. At higher apple polyphenol contents (1.2% and 1.6%), the effect of polyphenols on the protein gel network of processed cheese could lead to a decrease in the viscosity of the material, resulting in a poor forming effect and a rougher shape; in these cases, the porosity was 10.6% and11.1%, respectively. These samples showed gaps, indicating that the ductility was poor when the filaments were extruded, and broken lines appeared, causing the upper layers to be affected by the lower layers, which is consistent with the rheological properties of the samples.

The effect of apple polyphenols on the color of the 3D-printed processed cheese samples is shown in Table 2. The addition of apple polyphenols had a significant effect on the sample color (*p* < 0.05). Specifically, apple polyphenol addition resulted in lower L* and b* values but higher a* values, indicating that the samples became less bright, darker, and more yellow in color, and these changes were positively correlated to the polyphenol content.

### 3.5. Sensory Evaluation

Figure 6 shows the effect of different contents of apple polyphenols on the sensory quality of 3D-printed processed cheese products. According to the product’s color score, the appearance score of the product with 1.6% apple polyphenol content is the lowest, while the color score of the product with 0.4% apple polyphenol content is the highest, which is consistent with the results of the color analysis. According to the appearance score, the product with 0.8% apple polyphenol content has the highest appearance score, while the product with 1.6% apple polyphenol content has the lowest appearance score, which is consistent with the results of the size characteristics analysis. According to the flavor score, the product with 0.8% apple polyphenol content has the highest flavor score, but there is no significant difference in flavor compared to the product with 0.4% apple polyphenol content. The product with 1.6% apple polyphenol content has the lowest flavor score; this is because the excessive addition of apple polyphenols will present a certain bitterness. The overall scores of the products with 0.4% and 0.8% apple polyphenol contents exceeded the critical value of consumer acceptance (7 points). However, because the appearance and flavor scores of the product with 0.8% apple polyphenol content are higher than those of the product with 0.4% apple polyphenol content, the product with 0.8% apple polyphenol content has a higher consumer acceptance.

## 4. Conclusions

This study applied apple polyphenols to the 3D printing of processed cheese, and explored the effects of different apple polyphenol addition amounts on the antioxidant and structural stability of 3D-printed processed cheese. The addition of apple polyphenols can effectively improve the antioxidant capacity of 3D-printed processed cheese. When the content of apple polyphenols is greater than 0.4%, the ABTS free radical scavenging rate is above 60%, and the DHHP free radical scavenging rate is above 70%. At the same time, polyphenol molecules change the secondary structure of the protein in processed cheese by binding to it, and excessive addition of apple polyphenols will cause the viscosity of the processed cheese to continue to decrease and the gel properties to deteriorate, affecting the printing effect of the product. When the amount of added apple polyphenols is 0.8%, the 3D printing effect is optimal with a porosity rate of 4.1%. Overall, when the content of apple polyphenols is 0.8%, the resulting processed cheese can have both good antioxidant capacity and structural stability for 3D printing. This provides a new way for personalized antioxidant functional food.

## Figures and Tables

**Figure 1 foods-12-01731-f001:**
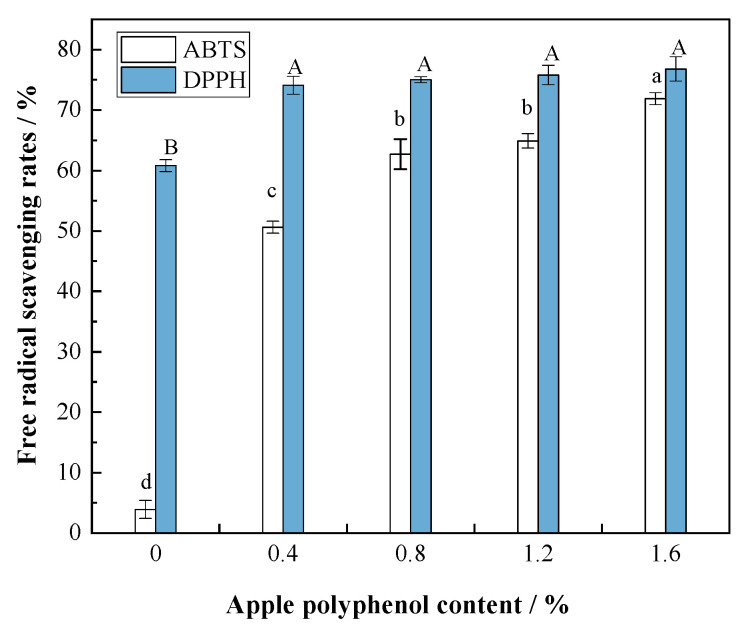
Effect of apple polyphenol addition on the free radical scavenging rates (%) of 2,2-di(4-*tert*-octylphenyl)-1-picrylhydrazyl (DPPH) and 2,2′-azinobis-(3-ethylbenzothiazoline-6-sulfonic acid) (ABTS) in processed cheese. Different letters in the figure indicate significant differences between values (*p* < 0.05).

**Figure 2 foods-12-01731-f002:**
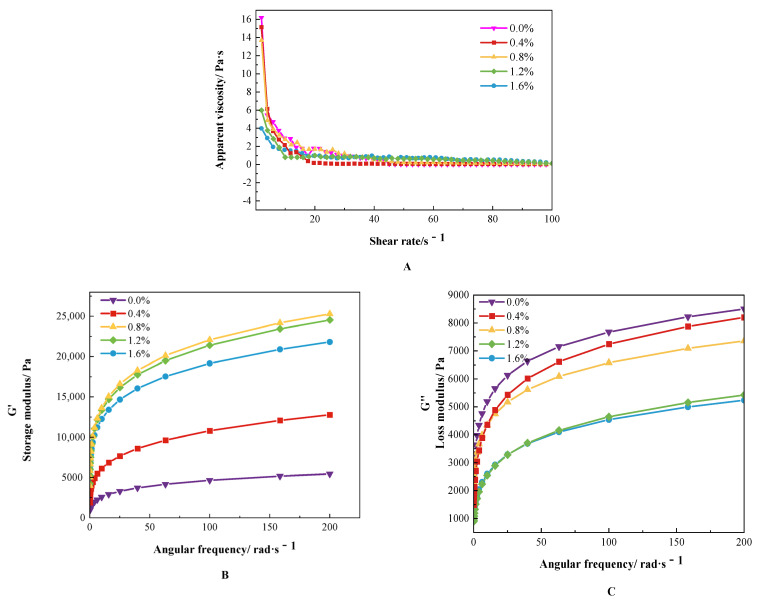
(**A**) Effect of added apple polyphenols on the apparent viscosity curves of processed cheese; (**B**,**C**) Effect of added apple polyphenols on the dynamic rheological properties of processed cheese: (**B**) storage modulus (G′) and (**C**) loss modulus (G″).

**Figure 3 foods-12-01731-f003:**
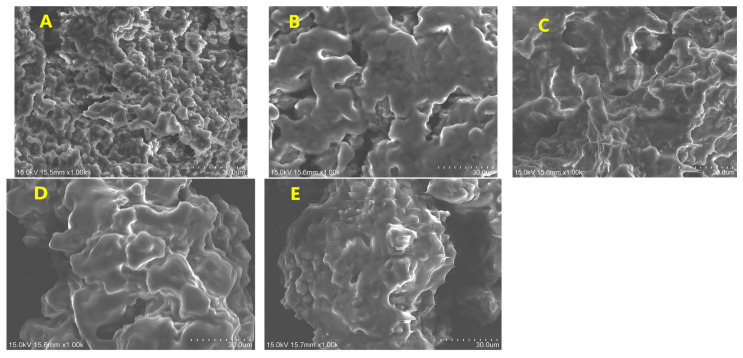
Scanning electron micrographs of processed cheese samples with (**A**) 0.0%, (**B**) 0.4%, (**C**) 0.8%, (**D**) 1.2%, and (**E**) 1.6% apple polyphenols.

**Figure 4 foods-12-01731-f004:**
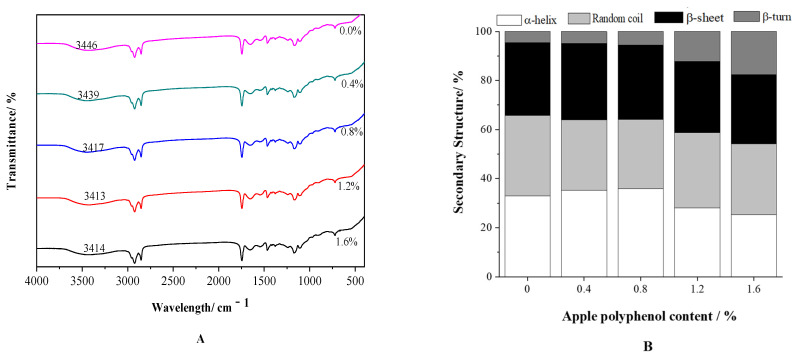
(**A**) Fourier transform infrared (FT-IR) spectra of processed cheese samples with added apple polyphenols; (**B**) Effect of apple polyphenol addition on the protein secondary structure of processed cheese.

**Figure 5 foods-12-01731-f005:**
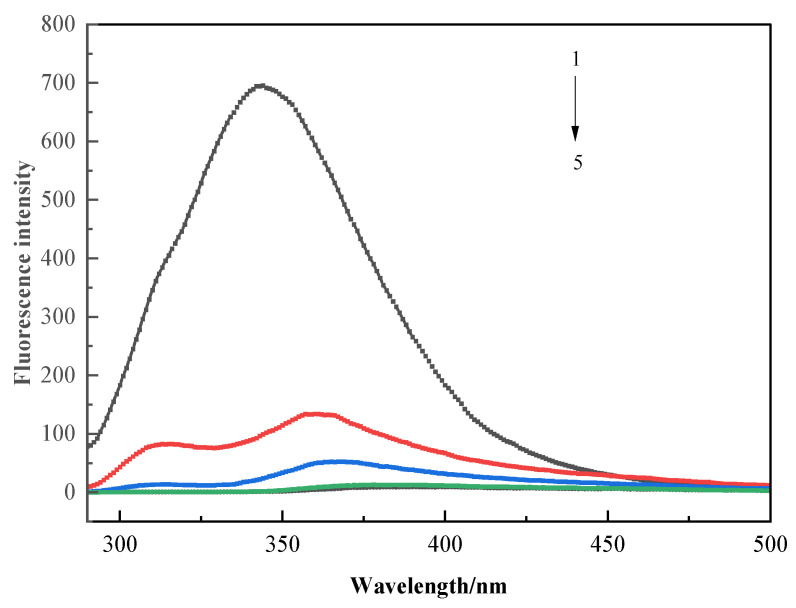
Effect of apple polyphenol addition on the fluorescence spectra of processed cheese. Labels 1 to 5 indicate apple polyphenol content of 0% to 1.6%.

**Figure 6 foods-12-01731-f006:**
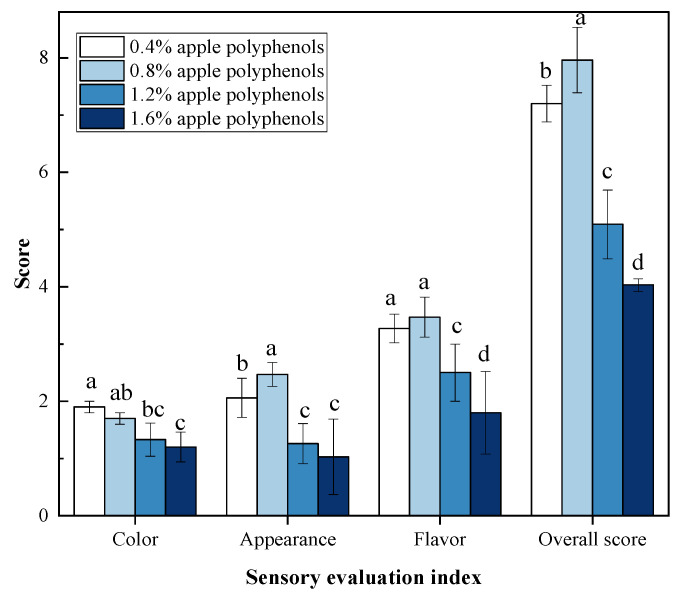
Effect of different apple polyphenol contents on the sensory evaluation of 3D-printed processed cheese samples. Different letters in the figure indicate significant differences between values (*p* < 0.05).

**Table 1 foods-12-01731-t001:** Size characteristics and molding effects of 3D-printed processed cheese with different apple polyphenol content.

Addition (%)	Image	Length (mm)	Width (mm)	Height (mm)	Porosity (%)
0.0	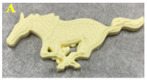	48.13 ± 0.01 ^c^	27.96 ± 0.05 ^b^	2.95 ± 0.01 ^a^	4.16 ± 0.06 ^b^
0.4	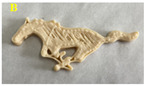	48.22 ± 0.03 ^a^	28.33 ± 0.21 ^b^	2.95 ± 0.02 ^a^	4.23 ± 0.08 ^b^
0.8	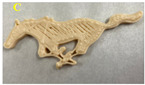	48.12 ± 0.02 ^c^	27.93 ± 0.06 ^b^	2.95 ± 0.03 ^a^	4.1 ± 0.10 ^b^
1.2	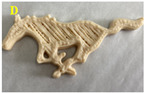	48.15 ± 0.05 ^b^	29.67 ± 0.13 ^a^	2.92 ± 0.03 ^a^	10.6 ± 0.46 ^a^
1.6	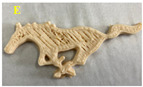	48.17 ± 0.08 ^b^	30.33 ± 0.58 ^a^	2.93 ± 0.02 ^a^	11.1 ± 0.77 ^a^

Different letters in the same line indicate significant differences between values (*p* < 0.05).

**Table 2 foods-12-01731-t002:** Color analysis of 3D-printed processed cheese with different apple polyphenol content.

Addition (%)	L*	a*	b*
0.0	84.69 ± 0.03 ^a^	−1.12 ± 0.01 ^e^	9.56 ± 0.01 ^b^
0.4	62.03 ± 0.02 ^b^	−0.08 ± 0.02 ^d^	12.89 ± 0.01 ^a^
0.8	55.03 ± 0.01 ^c^	0.21 ± 0.01 ^c^	8.71 ± 0.01 ^d^
1.2	52.02 ± 0.01 ^d^	0.51 ± 0.01 ^b^	8.52 ± 0.02 ^e^
1.6	50.81 ± 0.01 ^e^	1.35 ± 0.01 ^a^	8.84 ± 0.00 ^c^

Different letters in the same line indicate significant differences between values (*p* < 0.05). The L* value is the brightness, the a* value is the redness and greenness, and the b* value is the yellowness and blueness.

## Data Availability

No new data were created or analyzed in this study. Data sharing in not applicable to this article.

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
