# Peer review of "Effect of Apple Polyphenols on the Antioxidant Activity and Structure of Three-Dimensional Printed Processed Cheese"

_foods, 2023, doi:10.3390/foods12081731_

Round 1

Reviewer 1 Report

After careful examination of the manuscript “Effect of Apple Polyphenols on the Antioxidant Activity and Structure of Three-Dimensional Printed Processed Cheese” I must admit that the idea is quite good, however, performance and final results leave a lot to be desired. The first main problem is linked with the lack of the statistic. The results do not have any bars of error which means that the results cannot be comparable and some of the results can even be over-interpreted. It is clear that the addition of phenolic compounds from apple powder will increase the antioxidant activities of processed cheese samples, but the main question is the reason why they were added. So much more interesting would be to know do they increase the oxidative stability of cheese samples. Unfortunately, such analyses were not performed. Another issue is linked to the quality of obtained products which can be seen in Figure 6. It can be seen that the addition of apple powder affects the quality of the final product. It would be very important to know how the customers will react to this aspect. Only by visual observation the control sample is the most attractive in my opinion. Another issue is taste, therefore; the sensory analysis in this study would give benefits as well. Finally, the overall scientific quality of the present manuscript is low, therefore I cannot advise the present manuscript for publication.

Author Response

Dear reviewer:

We greatly appreciate your valuable feedback, which has greatly helped to improve the quality of our paper. We have carefully revised our manuscript based on their suggestions, as well as the revisions requested by the editorial office. Please refer to the "response letter" copied below for our detailed response to all the points raised by the reviewers.

In addition, all the changes made to the manuscript have been highlighted in yellow in the files named "Manuscript highlighting the changes". If you have any further questions, please do not hesitate to contact me. I will respond as soon as possible.

Once again, thank you for your feedback and guidance.

Best regards,

Prof. Gongnian Xiao

Reviewer(s)' Comments to Author:

Reviwer1:

After careful examination of the manuscript “Effect of Apple Polyphenols on the Antioxidant Activity and Structure of Three-Dimensional Printed Processed Cheese” I must admit that the idea is quite good, however, performance and final results leave a lot to be desired.

1.The first main problem is linked with the lack of the statistic. The results do not have any bars of error which means that the results cannot be comparable and some of the results can even be over-interpreted. It is clear that the addition of phenolic compounds from apple powder will increase the antioxidant activities of processed cheese samples, but the main question is the reason why they were added. So much more interesting would be to know do they increase the oxidative stability of cheese samples. Unfortunately, such analyses were not performed.

Response: Thank you for reviewing our manuscript. We greatly appreciate your feedback and agree that there are still areas for improvement. Firstly, we have supplemented the error analysis and relevant data statistics (as shown in Figure 1, Table 1, Table 2, and Figure 6). The main reasons for adding apple polyphenols are explained in the introduction. Thank you for your suggestions on how to improve the manuscript.

2.Another issue is linked to the quality of obtained products which can be seen in Figure 6. It can be seen that the addition of apple powder affects the quality of the final product. It would be very important to know how the customers will react to this aspect. Only by visual observation the control sample is the most attractive in my opinion. Another issue is taste, therefore; the sensory analysis in this study would give benefits as well. Finally, the overall scientific quality of the present manuscript is low, therefore I cannot advise the present manuscript for publication.

Response:Thank you for your feedback. I agree that the quality of the final product is affected by the addition of apple powder, and it would be important to consider how customers will react to this aspect. I also agree that sensory analysis would provide valuable information regarding taste. Therefore, we have added sensory analysis, as shown in Figure 6 Effect of different apple polyphenol contents on the sensory evaluation of 3D printed processed cheese samples.

Reviewer 2 Report

The work entitled “Effect of Apple Polyphenols on the Antioxidant Activity and Structure of Three-Dimensional Printed Processed Cheese” addresses a novel topic in the field of food technology. It is of scientific interest and also is significant for the industrial practice.

The study design is adequate. However, I believe that the interest of the work should be more clearly stated in the introduction. Justify why it is necessary to improve the structure in this product.

Additionally, the methodology used for the analysis should be described more precisely. Finally, a section should be included with the main conclusions of the work focusing on the most important discoveries.

Below my considerations line by line:

Abstract

-          - Line 16. Indicate the % of polyphenols used

Introduction

-         -  Introduce the concept of processed cheese: what does this term refer to?

Justify why it is necessary to improve the structure in this product.

- Line 47. Why do you refer to the interest of apple polyphenols? Could it be referring to polyphenols of another origin?. Could polyphenols from another origin have this same effect?

-          - Line 52. Cheese? Or the formulation of proccesed cheese?

-        -   Lines 58-61. Don´t anticipate the results of the study in the introduction.

If it refers to previous studies, it is necessary to indicate the references.

Materials and Methods

-         - Indicate how many replicates of each analysis were performed

-         -  Line 65 and 71: Anjia or Anja?

-         -  Line 73. How was homogenization carried out?

-          - Line 77. Review terms “sterilised at 85ºC” and “refrigerated at -4ºC”.

-          - Indicate the reference on which the DPPH method used is based

-          - Line 84. Describe how the sample is prepared for the determination

-          - Line 84. How is incubation done? In what conditions?

-          - Indicate the reference on which the Test Kit method used is based

-        -   Line 93. Describe the conditions of the sample and how the extract is prepared. Describe how homogenization is performed.

-        -   Line 100. Indicate complete reference of the rheometer

-         -  Line 109 and 110. Processed cheese instead of cheese

-          - Line 110. Liofilizacion instead of Vacuum freeze-drying

-         -  Line 112. Indicate complete reference of the FT-IR spectrometer.

-          - Line 115. Indicate complete reference of the fluorescence spectrometer.

-         -  Line 117. PBS indicate with letters

-         -  Line 121. Indicate complete reference of the microscope

-         -  Line 123. Liofilizacion instead of Vacuum freeze-drying

-          - Line 128. Indicate complete reference of the printer

-          - Line 133. Origin 9.1. indicate the meaning

-          What were the criteria used to select the optimal conditions for printing?

Results and Discussion

-         -  Line 139. According to the data shown, the results are not directly proportional

-         -  Line 146. According to the data shown, the results are not directly proportional

-          - Line 151-153. Do these results mean that both determinations should be considered to determine the antioxidant capacity? Explain properly.

-          - Figure 1. Leyend: Free radical scavenging rates. Add (%).

Add At the bottom of the figure 1 the legend of the x axis: Free radical scavenging rates (%)

-          - Figure 2. It would be convenient to use in the three figures the same symbols and colors for each concentration of polyphenols

-         -  Lines 174-175. Indicate the values up to which it increases

-          - Line 181. Indicate an example of these data where it can be seen that G' is greater than G''

-         -  Line 233. Apparently, not much of a change from 0 to 0.8% polyphenols added. Is this difference significant?. Can you explain it, please?

-       -   Line 255. What do you mean by the expression "the composition was similar". Wasn't the composition always similar?

-         -  Line 258. Was the print better with 0.8% polyphenols than with 0%?

-         -  Line 265. The determination of colour must be included in the Materials and Methods section.

- In view of all the data, which concentration offers the best results? No overall assessment is made (Discussion).

Conclusions

There is no section of conclusions.

A section should be included with the main conclusions of the work, focusing on the most important discoveries.

However, this aspect has been included in the abstract.

Author Response

Dear reviewer:

We greatly appreciate your valuable feedback, which has greatly helped to improve the quality of our paper. We have carefully revised our manuscript based on their suggestions, as well as the revisions requested by the editorial office. Please refer to the "response letter" copied below for our detailed response to all the points raised by the reviewers.

In addition, all the changes made to the manuscript have been highlighted in yellow in the files named "Manuscript highlighting the changes". If you have any further questions, please do not hesitate to contact me. I will respond as soon as possible.

Once again, thank you for your feedback and guidance.

Best regards,

Prof. Gongnian Xiao

Reviewer(s)' Comments to Author:

Reviwer2:

The work entitled “Effect of Apple Polyphenols on the Antioxidant Activity and Structure of Three-Dimensional Printed Processed Cheese” addresses a novel topic in the field of food technology. It is of scientific interest and also is significant for the industrial practice.

Response: Thank you very much for your positive feedback.

  1. The study design is adequate. However, I believe that the interest of the work should be more clearly stated in the introduction. Justify why it is necessary to improve the structure in this product.

Response: Thank you very much for your suggestion. We have made modifications in the introduction, related research reports that apple polyphenols can also form complex gels with protein macromolecules, but their structure may be affected by apple polyphenols, thereby affecting the properties and functions of the gel [9]. For example, Zhou et al. [10] found that apple polyphenols have a significant impact on the gel properties of whey protein isolate, and moderate addition of apple polyphenols can effectively improve its structural characteristics. It is worth noting that the gel properties of food are also an important factor affecting the 3D printing process. Therefore, while fully utilizing the antioxidant advantages of apple polyphenols, it is also necessary to further understand the influence of apple polyphenols on the gel structure of food and its application value.

  1. Additionally, the methodology used for the analysis should be described more precisely. Finally, a section should be included with the main conclusions of the work focusing on the most important discoveries.

Below my considerations line by line:

Abstract

-   - Line 16. Indicate the % of polyphenols used.

Response: The  % of polyphenols has been increased(0%,0.4%,0.8%,1.2%,1.6%).

Introduction

- -  Introduce the concept of processed cheese: what does this term refer to?

Justify why it is necessary to improve the structure in this product.

Response: processed cheese is a homogeneous and easy-to-store dairy product made by heating and stirring different natural cheeses with added emulsifying salts and other dairy or non-dairy ingredients.

The addition of apple polyphenols may cause changes in the gel structure of the processed cheese, thus affecting the 3D printing performance of the final product. Therefore, it is necessary to regulate the amount of apple polyphenols added to ensure a good 3D molding effect based on improving the antioxidant activity of the product.

- - Line 47. Why do you refer to the interest of apple polyphenols? Could it be referring to polyphenols of another origin?. Could polyphenols from another origin have this same effect?

Response: Apple polyphenols have always been our research interest, referring to one source of polyphenols, and other sources of polyphenols have the same effect.

- - Line 52. Cheese? Or the formulation of proccesed cheese?

Response: Cheese can be broadly divided into natural cheese and processed cheese, therefore including processed cheese. processed cheese is processed from natural cheese as the main raw material.

- -   Lines 58-61. Don´t anticipate the results of the study in the introduction.

If it refers to previous studies, it is necessary to indicate the references.

Response: Thank you for your suggestion. It has been deleted from the manuscript.

Materials and Methods

- - Indicate how many replicates of each analysis were performed

Response: It has been noted that the analysis is repeated 3 times each time

- -  Line 65 and 71: Anjia or Anja?

Response: We have modified Anja to Anjia.

- -  Line 73. How was homogenization carried out?

Response: The mixture was placed under a high-speed homogenizer and homogenized intermittently at 6500 rpms for 2 minutes until uniform and non-grainy in appearance.

- - Line 77. Review terms “sterilised at 85ºC” and “refrigerated at -4ºC”.

Response: We have checked that their terms can be expressed as "sterilised at 85ºC" and "refrigerated at 4ºC". Terms "sterilized at 85 º C" and "refreshed at -4 º C" according to literature reports (Food Chemistry, 2023,402,134141; International Dairy Journal,2023,137,105504)

- - Indicate the reference on which the DPPH method used is based

Response: We have added references for the DPPH method(Reference 11. LWT Food Sci. Technol. 2021, 150, 112047)

- - Line 84. Describe how the sample is prepared for the determination

Response: Prepare a 10 mmol/L phosphate buffer solution with pH 7.4 and use it as the solvent to prepare a 1 mg/mL processed cheese sample solution.

- - Line 84. How is incubation done? In what conditions?

Response: We have changed “After invading for 1 h” to “at room temperature and protected from light for 1 h”.

- - Indicate the reference on which the Test Kit method used is based

- -  Line 93. Describe the conditions of the sample and how the extract is prepare

Response: The method for measuring antioxidant capacity has been improved and the reference literature has been inserted as follows: A 7 mmol/L ABTS free radical solution and a 2.45 mmol/L potassium persulfate solution were prepared and mixed in equal volumes. The mixture was placed in the dark for 12 hours and then diluted with anhydrous ethanol to obtain ABTS working solution with an absorbance of 0.70 ± 0.02 at 732 nm. A 1.0 mL sample solution of re-made cheese with different apple polyphenol concentrations of 1.0 mg/mL was mixed with 3.9 mL of ABTS working solution and reacted for 6 minutes at room temperature. Anhydrous ethanol was used as a blank instead of the sample solution, and the absorbance of the reaction sample was measured at 734 nm. The formula for calculating the ABTS cation free radical scavenging rate is as follows, with three parallel experiments conducted:

                         (2)

Where A1 is the absorbance value of the re-made cheese sample after reacting with ABTS, and A0 is the absorbance value of anhydrous ethanol instead of the sample solution after reacting with ABTS.

- -   Line 100. Indicate complete reference of the rheometer

Response: We have supplemented the complete instrument model and manufacturer (DHR-2, TA Instruments, USA)

- -  Line 109 and 110. Processed cheese instead of cheese

Response: We have modified cheese to processed cheese

- - Line 110. Liofilizacion instead of Vacuum freeze-drying

Response: We have changed Vacuum freeze-drying to liofilizacion.

- -  Line 112. Indicate complete reference of the FT-IR spectrometer.

Response: We have supplemented the complete instrument model and manufacturer (V70 IR Spertrometer, Bruker Germany)

- - Line 115. Indicate complete reference of the fluorescence spectrometer.

Response: We have supplemented the complete instrument model and manufacturer reference of the fluorescence spectrometer (F4500, Hitachi, Japan)

- -  Line 117. PBS indicate with letters

Response: We have changed PBS to Phosphate buffer solution (PBS)

- -  Line 146. Indicate complete reference of the microscope

Response: We have added the model and manufacturer (SU1510, Hitachi, Japan)

-  -  Line 123. Liofilizacion instead of Vacuum freeze-drying

Response: We have changed Vacuum freeze-drying to liofilizacion.

- - Line 128. Indicate complete reference of the printer

Response: Reference has been supplemented (Reference 16. Drying Technology. 2021, 39, 1196–1204)

- - Line 133. Origin 9.1. indicate the meaning

Response: Origin 9.1. refers to software for data statistics and plotting

-  What were the criteria used to select the optimal conditions for printing?

Response: The optimal parameter conditions were obtained through preliminary experimental mapping and are reflected in a separate section of the paper.

Results and Discussion

- -  Line 139. According to the data shown, the results are not directly proportional

Response: We have changed direct professional to positively correlated

- -  Line 146. According to the data shown, the results are not directly proportional

Response: We have changed direct professional to positively correlated

- - Line 151-153. Do these results mean that both determinations should be considered to determine the antioxidant capacity? Explain properly.

Response: Yes. There are various methods for evaluating the total antioxidant capacity due to differences in detection principles, reaction conditions, etc., and each method has its range of applicability and characteristics. Currently, no single method can comprehensively evaluate the total antioxidant capacity of a sample. In experiments, at least two methods are often required for simultaneous determination.

- - Figure 1. Leyend: Free radical scavenging rates. Add (%).

Response: We have added (%)

Add At the bottom of the figure 1 the legend of the x axis: Free radical scavenging rates (%)

Response: We have added Free radial scanning rates (%) and made modifications to the graph

- - Figure 2. It would be convenient to use in the three figures the same symbols and colors for each concentration of polyphenols

Response: The colors and symbols of each apple polyphenol concentration have been unified in Figures 2A, B, and C

-  -  Lines 174-175. Indicate the values up to which it increases

Response: It has been pointed out that when the polyphenol content in apples increases from 0% to 1.6%,

-  - Line 181. Indicate an example of these data where it can be seen that G' is greater than G''

Response: As shown in the example, When apple polyphenol content was 0.8%, 1.2% and 1.6%, the G′ values were always greater than the G′′ values.

- -  Line 233. Apparently, not much of a change from 0 to 0.8% polyphenols added. Is this difference significant?. Can you explain it, please?

Response: We have already changed "significantly" to " some extent". In addition, there was a certain degree of increase compared to the blank group, but it was not significant.

- -   Line 255. What do you mean by the expression "the composition was similar". Wasn't the composition always similar?

Response: They are similar, but among the 0.4%, 0.8%, 1.2%, and 1.6% polyphenols, the 0.8% had the best molding effect and was most similar to the blank group. It is difficult to compare based on image appearance alone. Therefore, as shown in Table 1, we have now supplemented the data analysis with size characteristics and porosity size characteristics.

- -  Line 258. Was the print better with 0.8% polyphenols than with 0%?

Response: Among 0.4%, 0.8%, 1.2%, and 1.6% polyphenols, 0.8% has the best printing effect and is most similar to 0%. To avoid ambiguity, the expression of the original sentence in the text has been changed to ”after the addition of apple polyphenols, the best 3D printing effect was achieved at an apple polyphenol content of 0.8%, and closest to the blank group print sample”.

- -  Line 265. The determination of colour must be included in the Materia

Response: The method of color analysis has been supplemented in the materials and methods

- In view of all the data, which concentration offers the best results? No overall assessment is made (Discussion).

Response: We are very sorry for our mistake. We have summarized the evaluation in the conclusion, polyphenol molecules change the secondary structure of the protein in processed cheese by binding to it, and excessive addition of apple polyphenols will cause the viscosity of the processed cheese to continue to decrease and the gel properties to deteriorate, affecting the printing effect of the product. When the amount of added apple polyphenols is 0.8%, the 3D printing effect is optimal with a porosity rate of 4.1%. In summary, when the concentration of apple polyphenols is 0.8%, the processed cheese can have both good antioxidant capacity and structural stability for 3D printing.

Conclusions

There is no section of conclusions.

A section should be included with the main conclusions of the work, focusing on the most important discoveries.

However, this aspect has been included in the abstract.

Response: We are very sorry for such a mistake. We have added the conclusion section to the manuscript and have proofread it in English.

Reviewer 3 Report

The manuscript aimed to explore the effects of different amounts of added apple polyphenols on the antioxidant properties and structure of 3D-printed processed cheese.

Please consider the following suggestions:

Reagents section – please include the reagents used for the assays performed.

Please insert the method you used for the obtaining of the extract used for the antioxidant capacity assays.

Table 1 – Please insert a caption at the bottom of the table explaining the abbreviations and the symbols used in the table.

I believe that a Conclusions section should be inserted in the manuscript.

Author Response

Dear reviewer:

We greatly appreciate your valuable feedback, which has greatly helped to improve the quality of our paper. We have carefully revised our manuscript based on their suggestions, as well as the revisions requested by the editorial office. Please refer to the "response letter" copied below for our detailed response to all the points raised by the reviewers.

In addition, all the changes made to the manuscript have been highlighted in yellow in the files named "Manuscript highlighting the changes". If you have any further questions, please do not hesitate to contact me. I will respond as soon as possible.

Once again, thank you for your feedback and guidance.

Best regards,

Prof. Gongnian Xiao

Reviewer(s)' Comments to Author:

Reviwer3:

The manuscript aimed to explore the effects of different amounts of added apple polyphenols on the antioxidant properties and structure of 3D-printed processed cheese.

Please consider the following suggestions:

1.Reagents section – please include the reagents used for the assays performed.

Response: Thank you very much for your suggestion. We have added reagents for analysis “Phosphate buffer solution (PBS) was obtained from Fly Clean Biotechnology Co, Ethanol and KBr were obtained from Shanghai Lingfeng Chemical Reagent Co, 2,2′-azinobis-(3-ethylbenzothiazoline-6-sulfonic acid) (ABTS) and 2,2-di(4-tert-octylphenyl)-1-picrylhydrazyl (DPPH) were obtained from Sangon Biotech(Shanghai)Co.”

2.Please insert the method you used for the obtaining of the extract used for the antioxidant capacity assays.

Response: The method for measuring antioxidant capacity has been improved and the reference literature has been inserted as follows:

A 7 mmol/L ABTS free radical solution and a 2.45 mmon/L potassium persulfate solution were prepared and mixed in equal volumes. The mixture was placed in the dark for 12 hours and then diluted with anhydrous ethanol to obtain ABTS working solution with an absorbance of 0.70 ± 0.02 at 732 nm. A 1.0 mL sample solution of processed cheese with different apple polyphenol concentrations of 1.0 mg/mL was mixed with 3.9 mL of ABTS working solution and reacted for 6 minutes at room temperature. Anhydrous ethanol was used as a blank instead of the sample solution, and the absorbance of the reaction sample was measured at 734 nm. The formula for calculating the ABTS cation free radical scavenging rate is as follows, with three parallel experiments conducted: ABTS free radic (%) = (1-A1/A0) × 100.

A1 is the absorbance value of the processed cheese sample after reacting with ABTS, and A0 is the absorbance value of anhydrous ethanol instead of the sample solution after reacting with ABTS.

3.Table 1 – Please insert a caption at the bottom of the table explaining the abbreviations and the symbols used in the table.

Response: We have inserted a title at the bottom of Table and explained the abbreviations and symbols used in the table.

4.I believe that a Conclusions section should be inserted in the manuscript.

Response: We are very sorry for such a mistake. We have added the conclusion section to the manuscript and have proofread it in English.

Round 2

Reviewer 1 Report

The manuscript has been improved, however, I am not convinced 100%.